# *Borrelia miyamotoi* a neglected tick-borne relapsing fever spirochete in Thailand

**Ratree Takhampunya**[1]*, **Asma Longkunan**[1], **Sakbuncha Somchaimongkol**[2], **Nittayaphon Youngdech**[1], **Nitima Chanarat**[1], **Jira Sakolvaree**[1], **Bousaraporn Tippayachai**[1], **Sommai Promsathaporn**[1], **Bhakdee Phanpheuch**[2], **Betty K. Poole-Smith**[1], **Patrick W. McCardle**[1], **Erica J. Lindroth**[1]

**1** Department of Entomology, United States Army Medical Directorate—Armed Forces Research Institute of Medical Sciences, Bangkok, Thailand, **2** Phop Phra Hospital, 245 Moo 2 Phop Phra District, Tak Province, Thailand

* Ratreet.fsn@afrims.org

**Data Availability Statement:** All Borrelia 16S rRNA and flagellin genes sequences are available from the GenBank (http://www.ncbi.nlm.nih.gov/Genbank/) database (accession number(s)

## Abstract

*Borrelia miyamotoi* is a relapsing fever spirochete that shares the same vector as Lyme disease causing *Borrelia*. This epidemiological study of *B. miyamotoi* was conducted in rodent reservoirs, tick vectors and human populations simultaneously. A total of 640 rodents and 43 ticks were collected from Phop Phra district, Tak province, Thailand. The prevalence rate for all *Borrelia* species was 2.3% and for *B. miyamotoi* was 1.1% in the rodent population, while the prevalence rate was quite high in ticks collected from rodents with an infection rate of 14.5% (95% CI: 6.3–27.6%). *Borrelia miyamotoi* was detected in *Ixodes granulatus* collected from *Mus caroli* and *Berylmys bowersi*, and was also detected in several rodent species (*Bandicota indica*, *Mus* spp., and *Leopoldamys sabanus*) that live in a cultivated land, increasing the risk of human exposure. Phylogenetic analysis revealed that the *B. miyamotoi* isolates detected in rodents and *I. granulatus* ticks in this study were similar to isolates detected in European countries. Further investigation was conducted to determine the serological reactivity to *B. miyamotoi* in human samples received from Phop Phra hospital, Tak province and in rodents captured from Phop Phra district using an in-house, direct enzyme-linked immunosorbent assay (ELISA) assay with *B. miyamotoi* recombinant glycerophosphodiester-phosphodiesterase (rGlpQ) protein as coated antigen. The results showed that 17.9% (15/84) of human patients and 9.0% (41/456) of captured rodents had serological reactivity to *B. miyamotoi* rGlpQ protein in the study area. While a low level of IgG antibody titers (100–200) was observed in the majority of seroreactive samples, higher titers (400–1,600) were also detected in both humans and rodents. This study provides the first evidence of *B. miyamotoi* exposure in human and rodent populations in Thailand and the possible roles of local rodent species and *Ixodes granulatus* tick in its enzootic transmission cycle in nature.

OP020595 to OP020610, and OP037786 to OP037794).

**Funding:** This work was funded by the Armed Forces Health Surveillance Branch, Global Emerging Infections Surveillance and Response System (AFHSB-GEIS), Silver Spring, Maryland, USA (P0100_21_AF to EJL). The funders had no role in study design, data collection and analysis, decision to publish, or preparation of the manuscript.

**Competing interests:** The authors have declared that no competing interests exist.

## Author summary

*Borrelia miyamotoi*, a causative agent of relapsing fever, has been recognized as a human pathogen worldwide. In Thailand, the spirochete bacteria was first detected in rodents collected from the far north in the country. The further investigation revealed its detection in several rodent species and *Ixodes granulatus* tick, indicating they could play a role in maintaining the enzootic transmission cycle of *B. miyamotoi* in nature. Epidemiological surveillance was conducted in Tak province where *B. miyamotoi* was previously detected in rodents. Although *B. miyamotoi* was not detected in human samples by molecular methods, the serological reactivity to *B. miyamotoi* recombinant protein in humans and rodents along with the detection of spirochete bacteria in rodents and ticks is a strong indication of *B. miyamotoi* transmission in this area. This is also the first report of *B. miyamotoi* serological reactivity in rodents and humans in Thailand; therefore, the possibility of *B. miyamotoi* infection should not be neglected. Understanding their epidemiological factors regarding their enzootic transmission, the risk of human exposure, and the awareness of disease prevalence will help control and prevent the disease.

## Introduction

*Borrelia miyamotoi*, a spirochete bacterium belonging to the relapsing fever (RF) group, has been described as a pathogen in humans [1–9]. It is carried by *Ixodes* hard ticks, the same genus transmitting *Borrelia burgdorferi* that causes Lyme disease [10–12]. *Borrelia miyamotoi* was initially detected in *Ixodes persulcatus* and *Apodemus* mice in Japan [13], and has subsequently been found in Europe and North America [10,14,15] where the distribution and prevalence of *Borrelia* infection are well documented. *Borrelia miyamotoi* has been phylogenetically classified into three types: Asian (transmitted mainly by *I. persulcatus*), European (*I. ricinus*) and American (*I. scapularis* and *I. pacificus*) [16]. Rodents could play important roles as reservoir hosts and several species have been reported to carry *B. miyamotoi*. For example, *Apodemus argenteus* from Japan [13] and *Peromyscus leucopus* from USA [10] were found to be naturally infected with the bacterium while *Myodes glareolus* from France [16], and *Apodemus flavicollis* from Hungary [17] were likely to play a role in the transmission cycle as well. Although the previous study detected *B. miyamotoi* in several rodent species (*Bandicota indica*, *Mus caroli*, and *Niviventer tenaster*) in Thailand [18], there has not yet been any study performed to confirm their roles as reservoir hosts in the maintenance of the bacterium in nature.

Human cases of *B. miyamotoi* were first reported in Russia [1], and later in the United States [19], Japan [7], and China [9], with the number of reported cases also increasing. However, few reports of this pathogen have come out of Southeast Asia in recent years. There were recent studies in Malaysia revealing that more than half of *I. granulatus* ticks were positive for *Borrelia* spp., related to *B. yangtzensis* [20]. Moreover, another report showed the Malaysian people had IgG antibodies against *B. burgdorferi* [21]. Interestingly, *B. miyamotoi* infected rodents were then reported for the first time in Malaysia [22]. In Thailand, our group previously reported that several *Borrelia* species were found in ticks and rodents distributed across the country [18]. *Borrrelia miyamotoi* was also detected in rodent tissues which had not previously been reported in Thailand [23]. The *B. miyamotoi* sequences found in Thailand were more closely related to genotypes from European countries than those found in Japan. Nevertheless, there has not yet been a report of a human case or vector in Thailand prior to our study. In this study, further investigation on the prevalence of *B. miyamotoi* in ticks and

rodents collected from Tak province (2019–2020), as well as in human patients (2018–2019) from the region was conducted. This is the first report of the serological reactivity to *B. miyamotoi* recombinant glycerophosphodiester-phosphodiesterase (rGlpQ) protein in both rodents and patients in Thailand, implying the possible human and rodent exposure to *B. miyamotoi* in the study area. These findings suggest that *Borrelia* spp. is widely present in Southeast Asia and likely occurs in other surrounding countries in addition to Thailand and Malaysia.

## Materials and methods

### Ethics statement

All sampling procedures and experimental manipulations were performed under an Institutional Animal Care and Use Committee (IACUC) approved protocol. Research was conducted in compliance with the Animal Welfare Act and other federal statutes and regulations related to animals and experiments involving animals, and adhered to principles outlined in the "Guide for the Care and Use of Laboratory Animals," NRC Publication, 2011 edition.

### Undifferentiated febrile illness (UFI) patients

A total of 84 individual UFI patients' coded specimens were received from Phrop Phra hospital, Tak province, Thailand (16˚24'04.8"N 98˚41'43.8"E). Samples were from outpatients and inpatients presenting to Phrop Phra hospital with UFI during 2018–2019. Residual whole blood and serum samples from routine laboratory testing were coded and sent to the Department of Entomology to be tested for the possible causative agent of UFI. The protocol was approved by Institute for the Development of Human Research Protections (IHRP), Health Systems Research Institute (HSRI), Thailand, on June 28, 2018. Since the work described herein involves the use of existing, coded specimens wherein investigators will not receive associated identifiable data, the project did not require a consent form or a review by the Institutional Review Board (IRB) and 45 CFR 46 and 32 CFR 219 does not apply.

### Rodent trapping and tick collection

Rodents and ticks were collected every other month in 2019 (February thru December) and in February, July, and September 2020 in Phop Phra district of Tak province, Thailand (16˚17'48.8"N 98˚40'31.9"E). All study sites were on private land, and permission was obtained from each of the owners to conduct research on their land. None of the field studies involved endangered or protected species. Rodents were captured using live traps baited with bananas, palm fruit, or dried fish, and were collected from orchards, palm and rubber plantations, cultivated rice-fields, grassland areas, edges of dense forest, stream margins, and around dwellings. All traps were set up in the evening between 1900 and 2200, and checked early in the morning. Traps were set for 3–5 nights for each collection. Traps were placed between 100–120 traps per night in the format of either grid or transect according to the topography. Trapping was organized at each collection site using several trap-lines and help from locals or the owner of the field for the information on the presence and the abundance of rats, their tracks, and locations of burrows in the edges of fields. Captured rodents were removed from the traps, euthanized using carbon dioxide, and processed immediately at the site of collection. Blood, serum, and tissue samples (liver, spleen, kidney, and lung) were collected and stored on dry ice. Ectoparasites, including ticks, were collected by combing or fine-tipped tweezers. Ears were removed and stored in 70% ethanol for chigger collection. All tissues were then transported to the Armed Forces Research Institute of Medical Sciences (AFRIMS) laboratory for further processing. All rodents were later identified to the species level as described by Muul et al. [24].

## Genomic DNA extraction

**UFI whole blood.** Genomic DNA was extracted from whole blood samples using an automated extraction machine, a QIAsymphony SP instrument (Qiagen, Hombrechtikon, Switzerland) with QIAsymphony DNA Mini Kit (Qiagen, Hilden, Germany). For each patient, 250 μl of whole blood was used for the DNA extraction using the DNA Blood 200 DSP protocol. DNA was eluted in 50 μl aliquots and stored at -20˚C until use. Ultrapure DNA/RNA-free distilled water was also included in every extraction procedure as an extraction control.

**Rodent tissue.** Spleen and kidney tissues from each rodent were cut into pieces (~3 mm in diameter) and added to 230 μl of ATL Tissue Lysis Buffer and 20 μl of Proteinase K solution (20 mg/ml), then incubated at 55˚C for 1 h or until the tissues were homogenized. A total volume of 250 μl homogenized solution was then used for DNA extraction on the QIAsymphony SP with QIAsymphony DNA Mini Kit and Tissue HC 200 DSP protocol. The DNA was eluted in 200 μl and stored at -20˚C until use. Ultrapure DNA/RNA-free distilled water was also included as an extraction control.

**Tick morphological identification and DNA extraction.** Ticks were identified morphologically [25–27] and pooled by animal host and tick species, life stage, and sex (pool size ranges from 1–4 ticks). Each pool was subjected to genomic DNA extraction using a modified protocol with the QIAamp DNA Mini Kit (Qiagen). Briefly, ticks in 180 μl of ATL buffer were punctured with a fine needle under a stereomicroscope to release the tissue from the hard exoskeleton prior to adding 20 μl of Proteinase K solution (20 mg/ml). Samples were then incubated at 55˚C for 1 h or until ticks were homogenized. A volume of 200 μl of AL buffer was added to the sample and the sample mixed and incubated at 70˚C for 10 min. Then 100 μl of absolute ethanol was added to precipitate DNA. The solution was transferred to a QIAamp DNA column then centrifuged at 8,000 rpm for 1 min. The supernatant was discarded. DNA was washed twice with 500 μl of QIAamp proprietary solutions AW1 and AW2, respectively. The DNA was eluted in 50 μl of AE buffer and stored at -20˚C until used. Ultrapure DNA/RNA-free distilled water was also included as an extraction control.

***Borrelia* spp. detection and species characterization.** Genomic DNA extracted from pooled ticks, rodent tissue, and patient's whole blood were screened for the presence of *Borrelia* using a genus-based TaqMan real-time PCR assay targeting the *Borrelia* 16S rRNA gene [28]. *Borrelia*-positive samples were further characterized by sequencing the flagellin (flaB) and 16S rRNA genes. The primer and probe sequences and conditions used in this study has been described previously by Takhampunya et al. [18]. The PCR products were purified using ExoSap-IT PCR Product Cleanup Reagent (Applied Biosystems, Foster City, CA) and sequenced using a SeqStudio genetic analyzer (Applied Biosystems). Raw sequences were edited and assembled using DNA Sequencher ver. 5.1 (Gene Code Corporation, Ann Arbor, MI).

**DNA sequence and phylogenetic analyses.** The pathogen sequences were aligned with reference sequences retrieved from the GenBank database using the MUSCLE codon alignment algorithm [29]. Maximum likelihood phylogenetic trees were constructed for each gene with the best fit nucleotide substitution model (flaB = Tamura—3 parameter + G model, 16S rRNA = Kimura—2 parameter + G model) using MEGA 6 software [30]. Bootstrap analyses with 1,000 resamplings were used to determine the node reliability of the tree branching.

**ELISA to *B. miyamotoi* rGlpQ.** To measure IgG directed against recombinant glycerophosphodiester-phosphodiesterase (rGlpQ) protein, an in-house enzyme-linked immunosorbent assay (ELISA) was performed following previously described protocols with minor modification [4,31]. Briefly, 96-well plate were coated overnight at 4˚C with 0.1 μg/well rGlpQ (BBI Solutions, Cardiff, UK) prepared in coating buffer. Next, the plate was washed three

times with 300 μl of washing buffer (0.1% Tween-20 in PBS). The plate was then incubated for 2 hours with blocking buffer (5% skim milk in washing buffer) at room temperature (RT). Each plate was subsequently washed three times with washing buffer and incubated for 1 hour at room temperature with 100 μl of diluted sample (1:100) in blocking buffer. After five times washes, 100 μl of either 1:1000 goat-anti rat IgG-peroxidase (SeraCare lifescience Inc., Milford, MA) or 1:3000 goat-anti human IgG-peroxidase (SeraCare lifescience Inc.) in blocking buffer was added and incubated further for 1 hour. Finally, each plate was washed three times with washing buffer and 100 μl of ABST 2-component microwell peroxidase substrate (SeraCare lifescience Inc.) was added and incubated at room temperature for 15–20 minutes in the dark. The optical density (OD) was read at 405 nm with a microplate reader (Molecular Devices, Sunnyvale, CA). Optical density values from non-coated well (without coated antigen) were used for background subtraction of corresponding samples. Sera obtained from 5 naïve Institute of Cancer Research (ICR) mice from a Charles River Technology (BioLASCO, Taiwan) or healthy blood donors were used as negative controls. The mean of the negative controls plus three times of standard deviation was used as the cutoff values for determining a positive result. Sample with OD value above 0.374 was considered positive/reactive and was further titered to determine their endpoint. The titer procedure was performed by diluting the positive sera by a factor of 2 (1:100, 1:200, 1:400, 1:800, 1:1600, and 1:3200) and tested again with the same procedure. The titer value was the inverse of the highest dilution with the OD greater 0.374.

**GlpQ Western blot antibody assay.**   Total 2 μg of rGlpQ (BBI Solutions) was subjected to separation by precast mini 4%–20% sodium dodecyl sulfate–polyacrylamide gel electrophoresis (Bio-Rad Laboratories, Hercules, CA). Protein was then transferred to PVDF membrane using Trans-Blot Turbo transfer system (Bio-Rad Laboratories). The membrane was blocked with PBS containing 10% skim-milk and 0.1% tween-20 for 2 hours at RT. Then, the membrane was separately incubated with 1:100 (5% skim-milk in PBST) of positive sample and normal human sera at 4˚C overnight. After washing five times (PBST), 1:5000 (5% skim-milk in PBST) of goat anti-human IgG-peroxidase (SeraCare lifescience Inc.) was added and incubated for 1 hour at RT. Protein bands were developed by TMB Enhanced One Component HRP Membrane Substrate (Sigma-Aldrich, St. Louis, MO) and reaction was stopped using deionized water. Finally, bands were observed using a Bio-RAD GelDoc XR Molecular Imager (Bio-Rad Laboratories).

## Data analysis

To estimate the probability of infection rate in pooled ticks, a maximum likelihood estimation (MLE) and a minimum infection rate (MIR) were calculated in Excel with the use of the CDC's Mosquito Surveillance Software tool (https://www.cdc.gov/westnile/resourcepages/mosqSurvSoft.html), which calculate the point estimate infection rate and 95% confidence intervals (CI) using pooled data that take into account individual pool samples sizes.

## Results

### Density and species distribution of rodents, small mammals, and their associated ticks collected in Phop Phra district, Tak province in 2019–2020

Surveillance was conducted in the same location in Wale sub-district, Phop Phra district every other month from February to December 2019 and in February, July, and September 2020. A total of 640 rodents and small mammals were collected, with the most abundant species being *Mus caroli* accounting for 41.6% (266/640), followed by *M. cookii* (21.3%, 136/640), and *M.*

*pahari* (10.3%, 66/640). Tick infestation on the captured rodents and small mammals appeared to be very low with a 0.1% average infestation rate (0–0.6%). Forty-three ticks were collected and 26 ticks were identified as *Ixodes granulatus* (15 females and 11 nymphs) and 17 ticks were *Haemaphysalis* spp. (10 nymphs and 7 larvae). The weather records from the nearest station to this area showed that the temperature and relative humidity were quite stable throughout the year and the rainy season covered the period from May to October (Fig 1A). The number of captured rodents peaked in the month of April and declined afterward, reaching the lowest point in October (Fig 1B). *Mus* rats were the most abundant species in this location and the abundance was relatively consistent throughout the year. The highest number of *M. caroli* began from Feb through June and the number declined sharply afterward (Aug-Oct) being replaced with other *Mus* species (*M. pahari* and *M. cookii*). Then the number of *M. caroli* increased again in late December. The densities of other rodent species such as *Rattus* rats and *Bandicota indica* were relatively low all year round when compared with *Mus* rats. Moreover, both species were captured during the dry season (Dec-Feb-Apr) more than the wet season.

## Prevalence of *Borrelia* spp. in humans, rodent population, and their associated ticks

The prevalence of *B. miyamotoi*, *B. theileri*, and *B. yangtzensis* in rodents and ticks collected in Phop Phra district, Tak province, Thailand in 2019–2020 is shown in Table 1. The collections were performed every other month (Feb-Dec) in 2019 and three collections were performed in 2020 (Feb, July, and September). Out of 640 animals, 15 were positive for *Borrelia* spp. (2.3% CI: 1.2–3.5%). The highest *Borrelia* spp. prevalence rate belongs to *Berylmys bowersi* (20.0% CI: -15.1–55.1%, 1/5) followed by *Leopoldamys sabanus* (12.5% CI:-10.4–35.4%, 1/8), and *B. indica* (11.8% CI: 0.9–22.6%, 4/34), whereas prevalence rates in *Mus* rats range from 0.7–7.1%.

*Borrelia* spp. was detected in ticks collected from captured rodents. Ticks were pooled from individual rodent hosts, then by tick species, and tick life stages before DNA extraction. Twenty-seven tick pools were screened for *Borrelia* spp. and the results show that a total of six pools were positive for *Borrelia* spp. (22.2%) with a maximum likelihood estimation (MLE) of 14.53% (95% CI: 6.28–27.64%) and a minimum infection rate (MIR) of 13.95% (95% CI: 3.60–24.31%); three pools (two pools of *I. granulatus* females and one pool of *Haemaphysalis* spp. nymphs) from *Mus caroli* (33.3%) had a MLE of 23.61% (95% CI: 6.80–52.70%) with a MIR of 21.43% (95% CI: 0–42.92%), two pools (*I. granulatus* females) from *Berylmys bowersi* (100%) had a MIR of 66.67% (95% CI: 13.32–120.01%) and one pool (*Haemaphysalis* spp. nymphs) from *Rattus tanezumi* (20.0%) had a MLE of 11.71% (95% CI: 0.75–44.03%) with a MIR of 12.50% (95% CI: 0–35.42%) Table 1.

All human samples (N = 84) were negative for a genus-based TaqMan real-time PCR assay targeting the *Borrelia* 16S rRNA gene (S1 Table).

## *Borrelia* species characterization using multigene analysis

*Borrelia* species characterization using multigene analysis (flagellin, 16S rRNA) was performed on all positive samples. DNA sequence and phylogenetic analyses revealed that there were three *Borrelia* species identified in this study; *Borrelia miyamotoi*, *Borrelia theileri*, and *Borrelia yangtzensis* (Figs 2 and 3). *Borrelia miyamotoi* was detected in seven rodents (7/640, 1.1%); *Bandicota indica* (3/34, 8.8%), *Mus* rats (3/482, 0.6%), and *Leopoldamy sabanus* (1/8, 12.5%) (Table 1). Their 16S rRNA and fla gene sequences fall within the clade of *B. miyamotoi* reference sequences with % identity ranging from 99.5–99.8 and 99.2–99.6, respectively (Table 2). *Borrelia theileri* was detected in two *Mus caroli* (2/266, 0.8%) and their 16S rRNA sequences were very similar to *B. theileri* reference sequences with % identity ranging from 99.6–100.

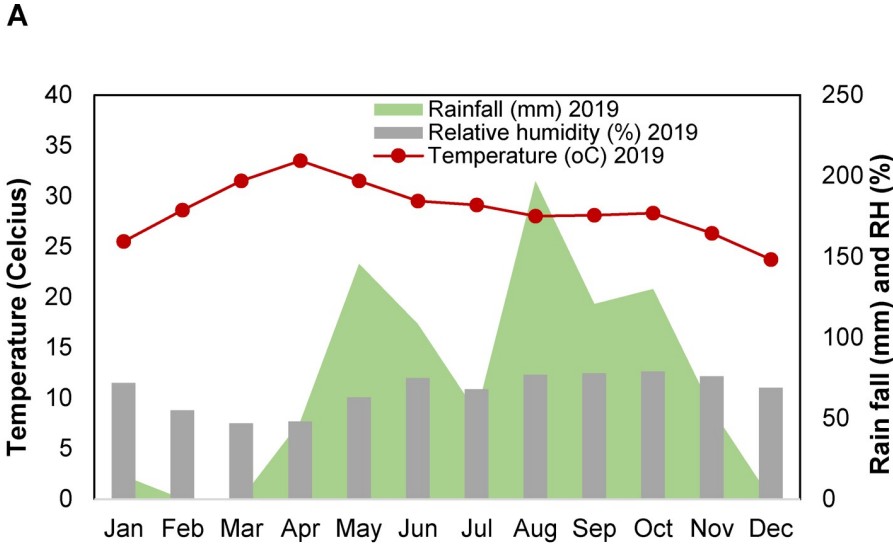

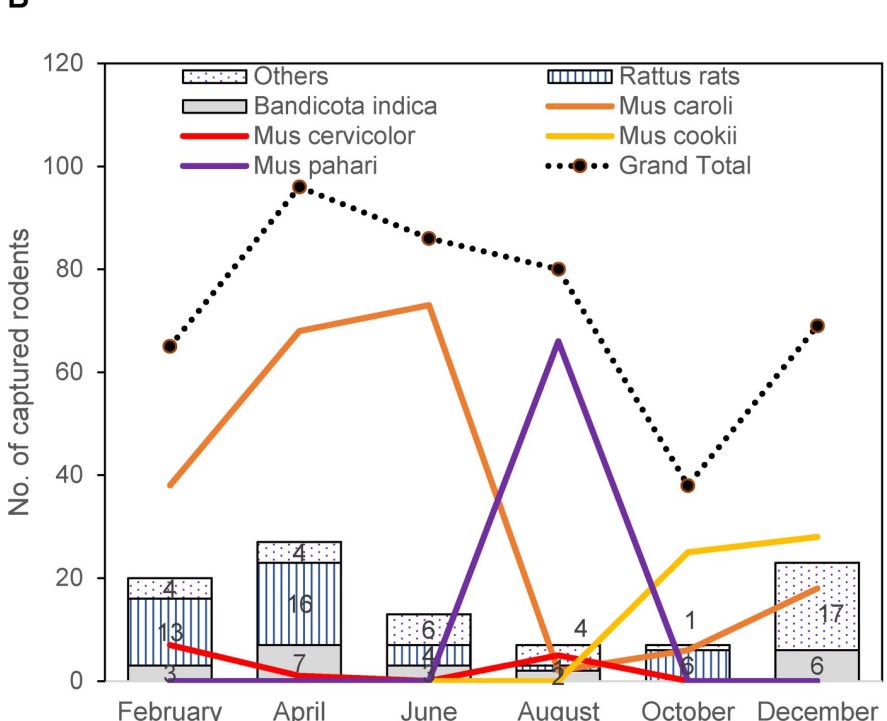

**Fig 1. Rainfall (mm) and temperature (˚C) during the year 2019 in Phop Phra district, Tak province in 2019 (A).**
The density and distribution of rodent species captured over one year period in Wale sub-district, Phop Phra district,
Tak province in 2019 (B).

*Borrelia yangtzensis* was detected in six rodents (6/640, 0.9%); *B. indica* (1/34, 2.9%), *M. caroli*
(1/266, 0.4%), *M. pahari* (1/66, 1.5%), *B. bowersi* (1/5, 20.0%), and *B. berdmorei* (2/27, 7.4%).
Their 16S rRNA sequences fall within the group of *B. yangtzensis* and *B. valaisiana* with %
identity ranging from 99.6–99.8. However, the fla gene sequences are somewhat distant from
those *Borrelia* species with a 98.0% identity to *B. yangtzensis* and a 96.8% to *B. valaisiana*.

**Table 1. prevalence in rodents and their associated ticks (2019–2020).** *Borrelia* spp.

| Host species | Rodents | | | | | Ticks | | | | | | | |
|---|---|---|---|---|---|---|---|---|---|---|---|---|---|
| | Number of captured rodents | B. miyamotoi-positive rodents (% Infection (95% CI)) | B. theileri-positive rodents (% Infection (95% CI)) | B. yangtzensis-positive rodents (% Infection (95% CI)) | All Borrelia-positive rodent (% Infection (95% CI)) | No. of Ixodes spp. | No. of Haemaphysalis spp. | Total number of collected ticks | No. of pool | Infestation rate | No. of Borrelia-positive tick pools (%) | MIR (95% CI) | MLE (95% CI) |
| *Bandicota indica* | 34 | 3 (8.8% (-0.7, 18.4%)) | 0 | 1 (2.9% (-2.7, 8.6%)) | 4 (11.8% (0.9, 22.6%)) | 11 | 1 | 12 | 5 | 0.4 | 0 | 0 | 0 |
| *Rattus exulans* | 34 | 0 | 0 | 0 | 0 | 0 | 0 | 0 | 0 | 0 | 0 | 0 | 0 |
| *Rattus tanezumi* | 22 | 0 | 0 | 0 | 0 | 1 | 7 | 8 | 5 | 0.4 | 1 (20.0%)¥ | 12.50 (0, 35.42) | 11.71 (0.75, 44.03) |
| *Mus caroli* | 266 | 1 (0.4% (-0.4, 1.1%)) | 2 (0.8% (-0.3, 1.8%)) | 1 (0.4% (-0.4, 1.1%)) | 4 (1.5% (0.0, 3.0%)) | 5 | 9 | 14 | 9 | 0.1 | 3 (33.3%)* | 21.43 (0, 42.92) | 23.61 (6.80, 52.70) |
| *Mus cervicolor* | 14 | 1 (7.1% (-6.4, 20.6%)) | 0 | 0 | 1 (7.1% (-6.4, 20.6%)) | 0 | 0 | 0 | 0 | 0 | 0 | 0 | 0 |
| *Mus cookii* | 136 | 1 (0.7% (-0.7, 2.2%)) | 0 | 0 | 1 (0.7% (-0.7, 2.2%)) | 4 | 0 | 4 | 4 | 0 | 0 | 0 | 0 |
| *Mus pahari* | 66 | 0 | 0 | 1 (1.5% (-1.4, 4.5%)) | 1 (1.5% (-1.4, 4.5%)) | 1 | 0 | 1 | 1 | 0 | 0 | 0 | 0 |
| *Berylmys bowersi* | 5 | 0 | 0 | 1 (20.0% (-15.1, 55.1%)) | 1 (20.0% (-15.1, 55.1%)) | 3 | 0 | 3 | 2 | 0.6 | 2 (100%)≠ | 66.67 (13.32, 120.01) | N/A |
| *Berylmys berdmorei* | 27 | 0 | 0 | 2 (7.4% (-2.5, 17.3%)) | 2 (7.4% (-2.5, 17.3%)) | 1 | 0 | 1 | 1 | 0 | 0 | 0 | 0 |
| *Leopoldamys sabanus* | 8 | 1 (12.5% (-10.4, 35.4%)) | 0 | 0 | 1 (12.5% (-10.4, 35.4%)) | 0 | 0 | 0 | 0 | 0 | 0 | 0 | 0 |
| *Others* | 28 | 0 | 0 | 0 | 0 | 0 | 0 | 0 | 0 | 0 | 0 | 0 | 0 |
| **Grand Total** | **640** | **7 (1.1% (0.3, 1.9%))** | **2 (0.3% (-0.1%, 0.8%))** | **6 (0.9% (0.2, 1.7%))** | **15 (2.3% (1.2, 3.5%))** | **26** | **17** | **43** | **27** | **0.1** | **6 (22.2%)** | **13.95 (3.60, 24.31)** | **14.53 (6.28, 27.64)** |

Note: ¥ , 1 Nymph *Haemaphysalis* sp. = *B. yangtzensis*; * , 1 Female *I. granulatus* = *Borrelia* sp., 1 Female *I. granulatus* = *B. miyamotoi*, 4 Nymphs *Haemaphysalis* sp. = *B. yangtzensis*; ≠ , 2 Females *I. granulatus* = *B. yangtzensis*, 1 Female *I. granulatus* = *B. miyamotoi*.

N/A = When all pools are positive, likelihood methods fail.

Others = *Berylmys mackenziei, Menetes berdmorei, Niviventer fulvescens, Tupaia belangeri, Vandeleuria oleracea*

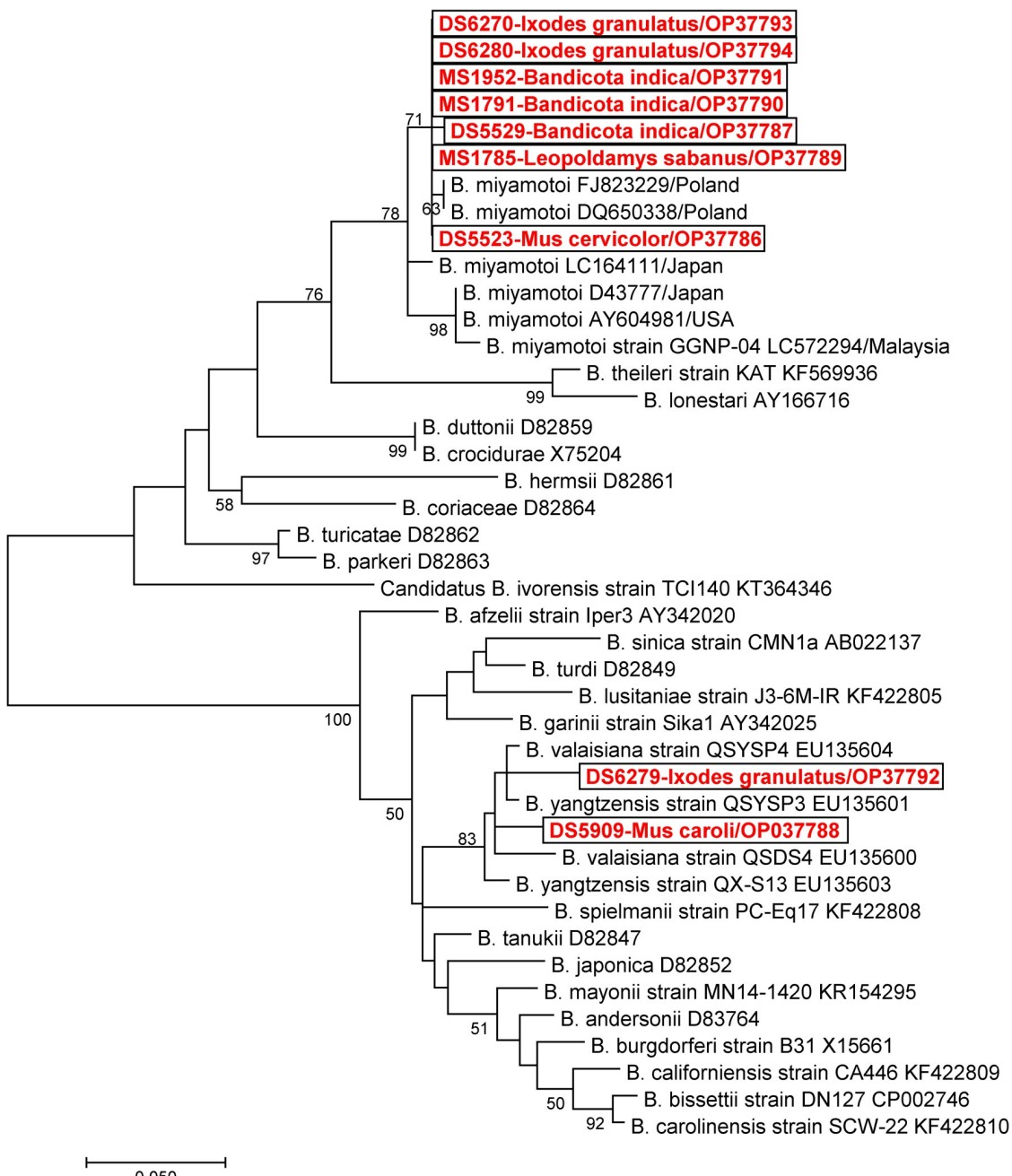

**Fig 2. Phylogenetic trees of 16S rRNA gene of *Borrelia* species detected in rodents and their associated ticks in Phop Phra district, Tak province, Thailand (2019–2020).**

Two pools of *I. granulatus* females collected from *M. caroli* and *B. bowersi* were positive for *B. miyamotoi* (2/27, 7.4%). Three tick pools (two pools of *Haemaphysalis* spp. nymphs and one pool of *I. granulatus* females) each collected from *R. tanezumi*, *M. caroli*, and *B. bowersi* were positive for *B. yangtzensis* (3/27, 11.1%) (Table 1). One *I. granulatus* female collected from *M. caroli* was positive for *B. theileri* (1/27, 3.7%).

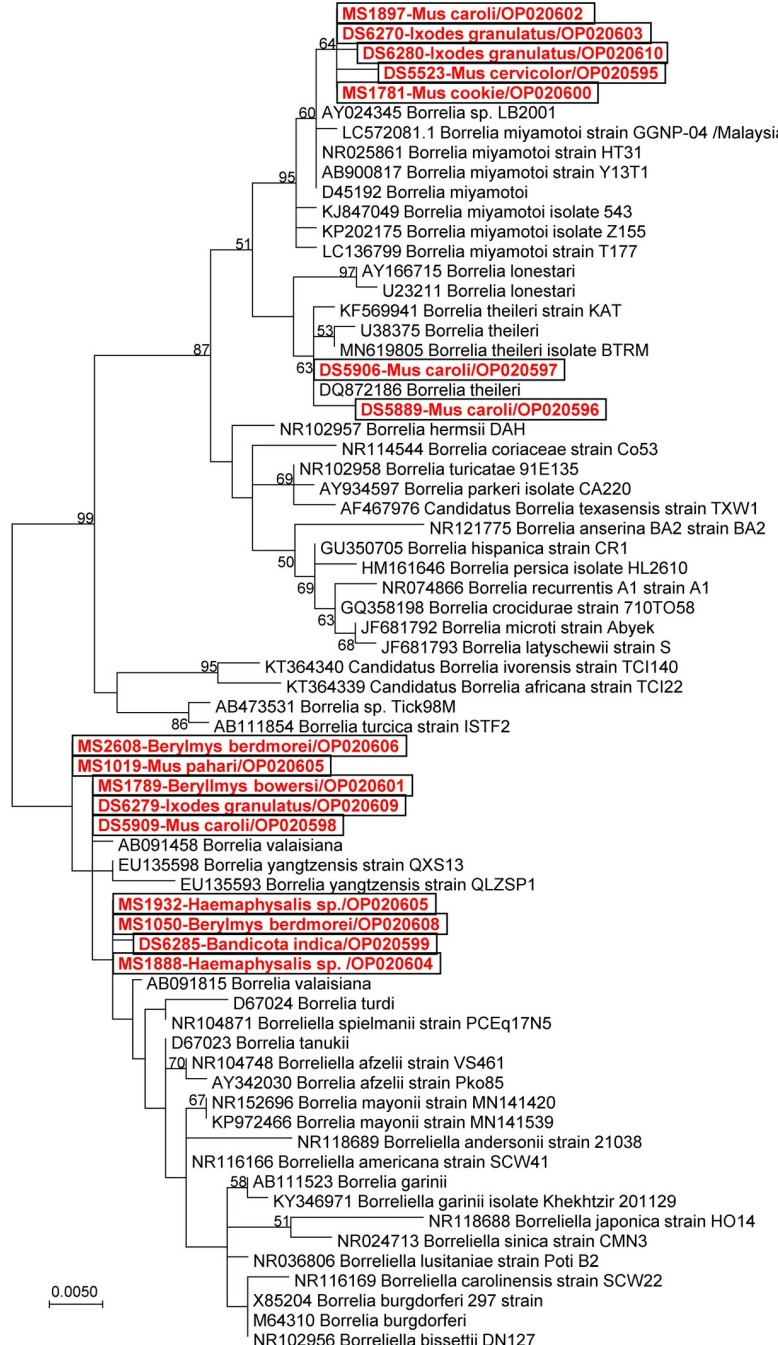

**Fig 3. Phylogenetic trees of flagellin gene of *Borrelia* species detected in rodents and their associated ticks in Phop Phra district, Tak province, Thailand (2019–2020).**

## Rodent exposure rate to *B. miyamotoi* and evidence of serological reactivity in humans

The serological reactivity to *B. miyamotoi* rGlpQ protein was determined in patient and rodent sera using the in-house ELISA assays detecting human IgG and rodent IgG against *B. miyamotoi* rGlpQ protein (38 kDa). The rGlpQ protein quality was checked by running the

**Table 2. Sequence identity matrix of *Borrelia* spp. detected in examined rodents and their associated ticks from Phop Phra District, Tak province (2019–2020).**

| Sequences/target gene(s) | B. miyamotoi | B. theileri | B. yangtzensis | B. valaisiana |
|---|---|---|---|---|
| **16S rRNA gene** | **NR025861 (Japan)** | **DQ872186** | **EU135598** | **AB091815** |
| DS5523-*Mus_cervicolor* | **99.5** | 98.3 | 95.9 | 96.1 |
| MS1781-*Mus_cookii* | **99.8** | 98.7 | 96.2 | 96.4 |
| MS1897-*Mus_caroli* | **99.8** | 98.7 | 96.2 | 96.4 |
| DS6270-*Ixodes_granulatus* (Tick) | **99.8** | 98.7 | 96.2 | 96.4 |
| DS6280-*Ixodes_granulatus* (Tick) | **99.6** | 98.5 | 96.0 | 96.2 |
| DS5889-*Mus_caroli* | 98.5 | **99.6** | 96.4 | 96.6 |
| DS5906-*Mus_caroli* | 98.9 | **100** | 96.8 | 97.0 |
| DS5909-*Mus_caroli* | 96.6 | 97.0 | **99.8** | 99.6 |
| MS1789-*Beryllmys_bowersi* | 96.6 | 97.0 | **99.8** | 99.6 |
| MS2608-*Berylmys_berdmorei* | 96.8 | 97.2 | **99.6** | 99.5 |
| MS1019-*Mus_pahari* | 96.8 | 97.2 | **99.6** | 99.5 |
| DS6279-*Ixodes_granulatus* (Tick) | 96.6 | 97.0 | **99.8** | 99.6 |
| DS6285-*Bandicota_indica* | 96.2 | 96.6 | 99.5 | **99.6** |
| MS1888-*Haemaphysalis*_sp. (Tick) | 96.4 | 96.8 | 99.6 | **99.8** |
| MS1932-*Haemaphysalis*_sp. (Tick) | 96.4 | 96.8 | 99.6 | **99.8** |
| MS1050-*Berylmys_berdmorei* | 96.4 | 96.8 | 99.6 | **99.8** |
| **Flagellin gene** | **DQ650338 (Poland)** | **KF569936** | **EU135601** | **EU135604** |
| DS5523-*Mus_cervicolor* | **99.6** | 89.9 | 77.9 | 77.5 |
| DS5529-*Bandicota_indica* | **99.2** | 89.4 | 77.4 | 76.9 |
| MS1785-*Leopoldamys_sabanus* | **99.6** | 89.9 | 77.9 | 77.5 |
| MS1791-*Bandicota_indica* | **99.6** | 89.9 | 77.9 | 77.5 |
| MS1952-*Bandicota_indica* | **99.6** | 89.9 | 77.9 | 77.5 |
| DS6270-*Ixodes_granulatus* (Tick) | **99.6** | 89.9 | 77.9 | 77.5 |
| DS6280-*Ixodes_granulatus* (Tick) | **99.6** | 89.9 | 77.9 | 77.5 |
| DS5909-*Mus_caroli* | 75.8 | 74.7 | **98.0** | 96.8 |
| DS6279-*Ixodes_granulatus* (Tick) | 77.0 | 75.9 | **96.8** | 96.4 |

recombinant protein on SDS-PAGE gel electrophoresis to ensure the 38 kDa recombinant protein was observed as expected (Fig 4A). ELISA assays were established and tested on all rodent and human samples included in this study. The specific binding of positive human serum (TAK016) to rGlpQ protein was shown as the expected 38 kDa band in a western blot assay (lane 2, Fig 4B), whereas a negative human serum has no band or signal presented (lane 1, Fig 4B). Furthermore, the cross-reactivity of the in-house ELISA assay was assessed against vector-borne diseases previously reported in the country [23] (Table 3). Positive controls (human IgG antibodies) from available commercial kits were used in this assessment. The results showed that at the initial serum dilution of 1:100 almost all positive controls had $OD_{405}$ values less than the cutoff value ($OD_{405}$ = 0.374) with the exception of a positive control for *Anaplasma phagocytophilum* ($OD_{405}$ = 0.601 ± 0.091). The positive control of *A. phagocytophilum* was further diluted (1:200–1:3200) and tested with the in-house ELISA assay again. The results showed that the positive control of *A. phagocytophilum* showed cross-reactivity to *B. miyamotoi* rGlpQ protein at the dilution of 1:100; however, the OD value was less than the cutoff value when it was diluted to 1:200 and the cross-reactivity was disappeared at dilutions of 1:400 to 1:3200.

Overall, 41 out of 456 rodents were seroreactive to *B. miyamotoi* rGlpQ protein accounting for 9.0% (95%CI: 6.4–11.6%) of all examined rodents. The highest percentage of positive anti-

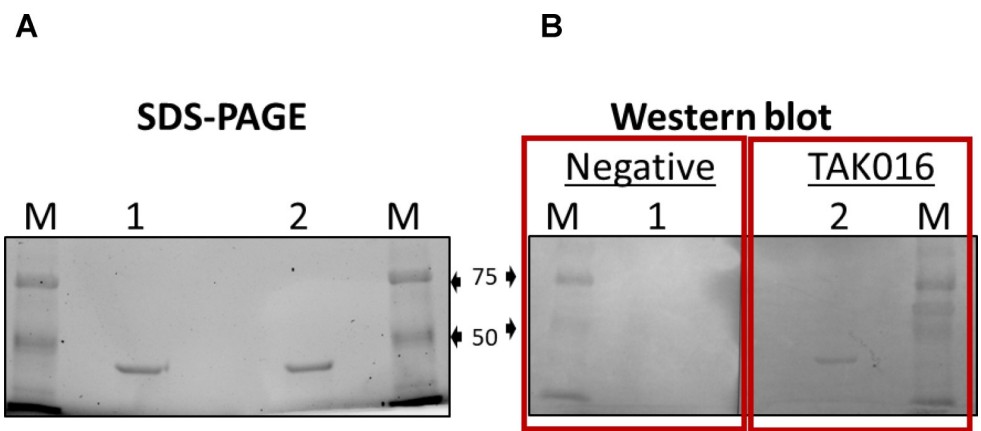

**Fig 4.** Polyacrylamide gel electrophoresis (SDS-PAGE) of *B. miaymotoi* recombinant GlpQ protein (A) and Western blot analysis (B) of Tak patient (TAK016) positive for *B. miyamotoi* by the in-house ELISA assay (titer 400).

*B. miyamotoi* rGlpQ IgG belongs to *Berylmys mackenziei* (37.5% CI: 4.0–71.0%, 3/8) followed by *Rattus* rats (29.3% CI: 15.3–43.2%, 12/41), *B. indica* (29.0% CI: 13.1–45.0%), and *B. berdmorei* (18.5% CI: 3.9–33.2%, 5/27) (Table 4). For *Mus* rats, an average of 3.7% (CI: 1.7–5.8%, 12/323) of positive anti-*B. miyamotoi* rGlpQ IgG was detected. The titer of IgG antibody to *B. miyamotoi* rGlpQ protein was determined using 2-fold serially diluted sera. The results showed that the majority of rodents (N = 20) had titers of 200, 10 had titers of 100, six had titers of 800, four had titers of 400, and only one had a titer of 1600. The percentage of positive anti-*B. miyamotoi* rGlpQ IgG in captured rodents was at a range of 1.6–9.1% throughout the year; however, the percentage sharply spiked in two months; April (19.3%), and September (14.8%) (Fig 5). Of note, the *B. miyamotoi* DNA was detected in rodents captured during the dry season.

Screening for human IgG antibody against *B. miyamotoi* rGlpQ protein was performed to determine the serological reactivity to *B. miyamotoi* in patients visiting Phop Phra hospital, Tak province where *B. miyamotoi* was also detected in rodents and ticks collected from the same district. A total of 84 patient sera were collected from this hospital during 2018–2019. The results show that 15 patients had serological reactivity to *B. miyamotoi* rGlpQ protein

**Table 3. The cross-reactivity of the in-house ELISA assay to vector-borne diseases previously reported in Thailand.**

| Pathogens | Company | Catalog Number | Optical Density (OD) values (Mean ± SD) for human IgG titers* | | | | | |
|---|---|---|---|---|---|---|---|---|
| | | | 100 | 200 | 400 | 800 | 1600 | 3200 |
| *Rickettsia typhi* | Fuller | RTG-96K | 0.067 ± 0.019 | - | - | - | - | - |
| Spotted fever group *Rickettsia* | Fuller | SFG-96K | 0.013 ± 0.006 | - | - | - | - | - |
| Scrub typhus (*Orientia tsutsugamushi*) | InBios | STGS-1 | 0.022 ± 0.015 | - | - | - | - | - |
| *Bartonella hensalae* | Fuller | BH-GC | 0.054 ± 0.034 | - | - | - | - | - |
| *Bartonella quintana* | Fuller | BQ-GC | 0.015 ± 0.006 | - | - | - | - | - |
| *Anaplasma phagocytophilum* | Fuller | EEG-120 | **0.601 ± 0.091** | 0.274 ± 0.054 | 0.133 ± 0.029 | 0.070 ± 0.018 | 0.038 ± 0.010 | 0.019 ± 0.007 |
| *Borrelia burgdorferi*# | Fuller | BBG-120 | 0.071 ± 0.016 | - | - | - | - | - |
| Healthy blood donors (N = 5) | | | 0.122 ± 0.084 | - | - | - | - | - |
| cut-off OD (mean+3SD) | | | **0.374** | - | - | - | - | - |

Note: *, OD405 values were from 4–6 replicates of 2–3 independent experiments; #, *Borrelia burgdorferi* has not been reported in Thailand but was included here to determine the cross-reactivity with the in-house ELSIA.

**Table 4. The serological reactivity to *B. miyamotoi* rGlpQ protein in rodents captured from Phop Phra district, Tak province (2019–2020).** The serological reactivity as well as rodent IgG antibody titers in each rodent species were shown.

| Rodent species | No. of rodent sera | No. of nonreactive rodents | No. of seroreactive rodents (% positive (95% CI)) | IgG antibody titer (%(95% CI)) | | | | |
|---|---|---|---|---|---|---|---|---|
| | | | | 100 | 200 | 400 | 800 | 1600 |
| *Bandicota indica* | 31 | 22 | 9 (29.0% (13.1, 45.0%)) | 1 (3.2% (-3.0, 9.5%)) | 6 (19.4% (5.5, 33.3%)) | 0 | 1 (3.2% (-3.0, 9.5%)) | 1 (3.2% (-3.0, 9.5%)) |
| *Berylmys berdmorei* | 27 | 22 | 5 (18.5% (3.9, 33.2%)) | 0 | 3 (11.1% (-0.7, 23.0%)) | 1 (3.7% (-3.4, 10.8%)) | 1 (3.7% (-3.4, 10.8%)) | 0 |
| *Berylmys mackenziei* | 8 | 5 | 3 (37.5% (4.0, 71.0%)) | 1 (12.5% (-10.4, 35.4%)) | 1 (12.5% (-10.4, 35.4%)) | 0 | 1 (12.5% (-10.4, 35.4%)) | 0 |
| *Mus caroli* | 130 | 123 | 7 (5.4% (1.5, 9.3%)) | 3 (2.3% (-0.3, 4.9%)) | 3 (2.3% (-0.3, 4.9%)) | 1 (0.8% (-0.7, 2.3%)) | 0 | 0 |
| *Mus cervicolor* | 7 | 7 | 0 | 0 | 0 | 0 | 0 | 0 |
| *Mus cookii* | 127 | 123 | 4 (3.2% (0.1, 6.2%)) | 1 (0.8% (-0.8, 2.3%)) | 2 (1.6% (-0.6, 3.7%)) | 1 (0.8% (-0.8, 2.3%)) | 0 | 0 |
| *Mus pahari* | 59 | 58 | 1 (1.7% (-1.6, 5.0%)) | 0 | 1 (1.7% (-1.6, 5.0%)) | 0 | 0 | 0 |
| *Rattus exulans* | 23 | 16 | 7 (30.4% (11.6, 49.2%)) | 4 (17.4% (1.9, 32.9%)) | 2 (8.7% (-2.8, 20.2%)) | 1 (4.4% (-4.0, 12.7%)) | 0 | 0 |
| *Rattus tanezumi* | 18 | 13 | 5 (27.8% (7.1, 48.5%)) | 0 | 2 (11.1% (-3.4, 25.6%)) | 0 | 3 (16.7% (-0.6, 33.9%)) | 0 |
| Others | 26 | 26 | 0 | 0 | 0 | 0 | 0 | 0 |
| **Grand Total** | **456** | **415** | **41 (9.0% (6.4, 11.6%))** | **10 (2.2% (0.9, 3.5%))** | **20 (4.4% (2.5, 6.3%))** | **4 (0.9% (0.0, 1.7%))** | **6 (1.3% (0.3, 2.4%))** | **1 (0.2% (-0.2, 0.7%))** |

accounting for 17.9% (CI: 9.7–26.0%) of all examined human samples. Four patients had titers of 400 (4.8% CI: 0.2–9.3%), five had titers of 200 (6.0% CI: 0.9–11.0%), and six had titers of 100 (7.1% CI: 1.6–12.7%). The highest number of patients with serological reactivity to BMD rGlpQ protein was from Phop Phra sub-district (7/22, 31.8% CI: 12.4–51.3%), followed by Khiri Rat sub-district (4/24, 16.7% CI: 1.8–31.6%), Wale sub-district (2/15, 13.3% CI: -3.9–30.5%), and Chong Khaep sub-district (1/8, 12.5% CI: -10.4–35.4%) (Table 5).

## Discussion

*Borrelia miyamotoi*, a causative agent of relapsing fever, has been recognized as a human pathogen worldwide, primarily in Russia, the United States, Europe, and central Asia. In Thailand, the spirochete bacteria was first detected in *Niviventer* rodents collected from the far north in the country [23]. Its distribution in other regions of the country was later reported in a retrospective study, which revealed its detection in several rodent species (*Bandicota indica*, *Mus caroli*, *Rattus surifer*, as well as *Niviventer tenaster*) which could potentially play a role in maintaining *B. miyamotoi* enzootic life cycle [18]. Prior to this study, *B. miyamotoi* had not been

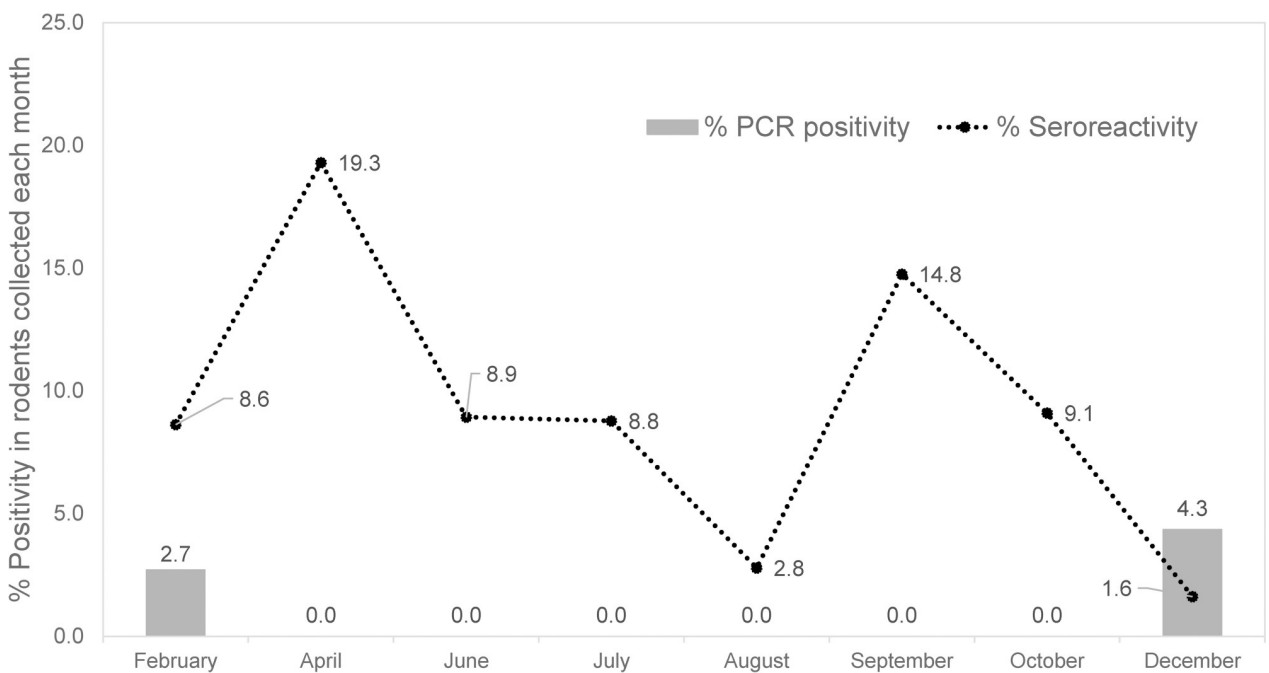

**Fig 5. The percentage of positive anti-*B. miyamotoi* rGlpQ IgG antibody in rodents collected every other month in 2019–2020, Phop Phra district, Tak province.**

detected in ticks in Thailand. This study is the first detection of *B. miyamotoi* in tick species, *Ixodes granulatus*, indicating this species could play a role in maintaining the enzootic transmission cycle of *B. miyamotoi* in nature. However according to Kahl et al. [32], the definition of the reservoir host and vector in the vector-borne transmission requires three key components 1) the pathogen has to be transmitted to the reservoir host via a vector bite, 2) pathogen must survive and multiply in the reservoir host, and 3) the pathogen must subsequently be acquired by a new vector upon biting the reservoir host and be transmitted to further hosts. The detection of *B. miyamotoi* in *I. granulatous* ticks collected from rodents in this study may

**Table 5. The serologial reactivity of human sera to *Borrelia miyamotoi* rGlpQ protein in Phop Phra district, Tak province, Thailand.**

| Sub-district | Number of human samples | Number of Serological reactivity (% (95% CI)) | Antibody titer (% (95% CI)) | | |
|---|---|---|---|---|---|
| | | | 100 | 200 | 400 |
| Others | 3 | 0 | 0 | 0 | 0 |
| Chong Khaep | 8 | 1 (12.5% (-10.4, 35.4%)) | 1 (12.5% (-10.4, 35.4%)) | 0 | 0 |
| Khiri Rat | 24 | 4 (16.7% (1.8, 31.6%)) | 3 (12.5% (-0.7, 25.7%)) | 0 | 1 (4.2% (-3.8, 12.2%)) |
| Phop Phra | 22 | 7 (31.8% (12.4, 51.3%)) | 1 (4.5% (-4.2, 13.2%)) | 3 (13.6% (-0.7, 28.0%)) | 3 (13.6% (-0.7, 28.0%)) |
| Ruam Thai Phatthana | 12 | 1 (8.3% (-7.3, 24.0%)) | 1 (8.3% (-7.3, 24.0%)) | 0 | 0 |
| Wale | 15 | 2 (13.3% (-3.9, 30.5%)) | 1 (6.7% (-6.0, 19.3%)) | 1 (6.7% (-6.0, 19.3%)) | 0 |
| **Total** | **84** | **15 (17.9% (9.7, 26.0%))** | **6 (7.1% (1.6, 12.7%))** | **5 (6.0% (0.9, 11.0%))** | **4 (4.8% (0.2, 9.3%))** |

or may not imply that tick or rodent species are vector or reservoir hosts for *B. miyamotoi*. In order to make this conclusion, studies of vector competence and reservoir host competence must be carried out experimentally. Epidemiological surveillance was conducted in Tak province where *B. miyamotoi* was previously detected in rodents from our previous study. Human whole blood samples were also received from the local district hospital (Phro Phra Hospital, Phop Phra District, Tak province). Although *B. miyamotoi* was not detected in human samples by molecular methods, the serological reactivity of human and rodent IgG antibodies against rGlpQ protein of *B. miyamotoi* along with the detection of spirochete bacteria in rodents and ticks is a strong indication of *B. miyamotoi* transmission in this area. No *Borrelia*-positive tick was collected from *Borrelia*-positive animals in this study. However, the detection of *Borrelia* spp. in feeding ticks does not necessarily imply the vectorial status of the tick species. Only a small number of ticks were collected from rodents in this area and attempts to collect questing ticks by dragging and flagging at several locations in this area including the rodent trapping sites were unsuccessful. More epidemiological studies and surveillance should be conducted in other areas in which *B. miyamotoi* has been detected such as Nan, Chiangrai, and Loei provinces [18]. This is also the first detection of *B. miyamotoi* serological reactivity in humans in Thailand; therefore, the possibility of *Borrelia* infection in humans should not be neglected. However, the cross-reaction to *B. miyamotoi* rGlpQ protein was observed with the positive control (human IgG) of *A. phagocytophilum* at the dilution of 1:100 but the cross-reactivity disappeared when it was diluted further (1:200–1:3200). Similar observation was reported by Jahfari et al. in the serologically unconfirmed but suspected human granulocytic anaplasmosis in the Netherlands [8]. Moreover, other groups also observed the low sensitivity and specificity of *B. miyamotoi* rGlpQ protein and suggested using other variable major proteins (Vmps) in conjunction with rGlpQ protein to improve serodiagnostic sensitivity [33–35]. In this current study, four human samples with high IgG antibody titer (400) were detected and the specific antigen-antibody binding was later confirmed by the Western blot assay. Potentially, the data presented in this study provide the serological evidence of *B. miyamotoi* exposure in humans in Thailand. The lack of information regarding the importance of noting tick bites among locals can lead to the underestimation by health care workers of the problem of tick and tick-borne disease infections among patients. Since borreliosis is not a reportable disease in Thailand, most patients with fever of unknown etiological agent would be reported as pyrexia of unknown origin (PUO). Patients with PUO very often would come with undifferentiated febrile illness and were not properly diagnosed and provided a treatment. Tak province was one of the top 5 highest PUO cases in the country in 2019 with a 1,785.21 morbidity rate per 100,000 population was reported (http://doe.moph.go.th/surdata/index.php). On the contrary, tick infestation is very well recognized in companion pets and livestock and insecticides for use by pet owners and farmers are widely accessible in the country.

In this study, *Borrelia* species detected in rodents and ticks are in accordance with previous findings in which *B. miyamotoi* and *B. yangtzensis* were found in equal proportions. Some *Borrelia* 16S rRNA sequences are equally similar to both *B. yangtzensis* and *B. valaisiana* (an agent causing Lyme disease), which makes species determination difficult without sequencing additional genes. However, since Lyme disease has never been found in Thailand, it is likely that the sequenced genes belong to *B. yangtzensis*. Genotypes of *B. miyamotoi* sequences detected in this study were more related to the genotypes from Europe than those from Asian countries (Japan and Malaysia), similar to the previous findings. The *Borrelia* spp. infection rate is a little higher in rodents than previously observed which could result from the frequency of sampling. The previous report was from one collection each in 2016 and 2017; however, in this study, the sampling was continuously conducted every other month for over one year. *Borrelia miyamotoi* was detected in three rodent genera similar to previous results as well as in the tick species,

*Ixodes granulatus*, the first detection in ticks in Thailand. These rodents inhabit cultivated areas such as paddy fields and different kinds of forested areas, which are close to rural communities. The detection of this newly described pathogen in rodents and ticks in cultivated lands has raised public health concerns among farmers working in the areas where the sylvatic transmission of *B. miyamotoi* occurs among vectors and reservoirs hosts.

The serological activity to *B. miyamotoi* rGlpQ protein in rodents strongly suggests that the bacterium causing BMD exists in the environment and poses a risk to humans. The evidence of *B. miyamotoi* sylvatic transmission and circulation in this province was clearly supported by the detection of the bacteria in rodents (*Mus* and *Bandicota* rats), and ticks, especially *I. granulatus*, collected from the same region and period of surveillance activity, and the serological reactivity in humans and rodents from the area.

In conclusion, *B. miyamotoi* disease is a neglected tick-borne disease and reports from past and current studies show that it is present in Thailand. The possibility that *I. granulatus* or other tick species may serve as vectors should be explored. More work is required to accurately evaluate the prevalence, range, and risk of human exposure, especially in areas such as forest habitats, mountain ranges, and national park where *I. granulatus* is predominantly found [36–39].

## Supporting information

**S1 Table. *Borrelia* spp. detection using a genus-based TaqMan real-time PCR assay targeting the *Borrelia* 16S rRNA gene in human samples received from Phop Phra hospital, Phop Phra district, Tak province, Thailand.**
(PDF)

## Acknowledgments

We are grateful to Mr. Taweesak Monkanna, Mr. Surachai Leepitakrat for their helps on fieldwork. A special thanks to all staffs at Phop Phra hospital for their excellent coordination during the study.

## Disclaimers

Material has been reviewed by the Walter Reed Army Institute of Research. There is no objection to its presentation and/or publication. The opinions or assertions contained herein are the private views of the author, and are not to be construed as official, or as reflecting true views of the Department of the Army or the Department of Defense. Research was conducted under an approved animal use protocol in an AAALACi accredited facility in compliance with the Animal Welfare Act and other federal statutes and regulations relating to animals and experiments involving animals and adheres to principles stated in the Guide for the Care and Use of Laboratory Animals, NRC Publication, 2011 edition.

## Author Contributions

**Conceptualization:** Ratree Takhampunya, Asma Longkunan.

**Data curation:** Ratree Takhampunya.

**Formal analysis:** Ratree Takhampunya.

**Funding acquisition:** Betty K. Poole-Smith, Patrick W. McCardle, Erica J. Lindroth.

**Investigation:** Ratree Takhampunya.

**Methodology:** Asma Longkunan, Nittayaphon Youngdech, Nitima Chanarat, Jira Sakolvaree, Bousaraporn Tippayachai, Sommai Promsathaporn.

**Resources:** Sakbuncha Somchaimongkol, Bhakdee Phanpheuch.

**Supervision:** Ratree Takhampunya.

**Visualization:** Ratree Takhampunya.

**Writing – original draft:** Ratree Takhampunya, Asma Longkunan.

**Writing – review & editing:** Erica J. Lindroth.

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
