## [Decision Letter · Decision Letter 0]

6 Nov 2022

Dear Dr. Takhampunya,

Thank you very much for submitting your manuscript "Borrelia miyamotoi a neglected tick-borne relapsing fever spirochete in Thailand." for consideration at PLOS Neglected Tropical Diseases. As with all papers reviewed by the journal, your manuscript was reviewed by members of the editorial board and by several independent reviewers. In light of the reviews (below this email), we would like to invite the resubmission of a significantly-revised version that takes into account the reviewers' comments. 

We cannot make any decision about publication until we have seen the revised manuscript and your response to the reviewers' comments. Your revised manuscript is also likely to be sent to reviewers for further evaluation.

Sincerely,

Wen-Ping Guo

Academic Editor

Mathieu Picardeau

Section Editor

Reviewer's Responses to Questions

**Key Review Criteria Required for Acceptance?**

**Methods**

-Are the objectives of the study clearly articulated with a clear testable hypothesis stated?

-Is the study design appropriate to address the stated objectives?

-Is the population clearly described and appropriate for the hypothesis being tested?

-Is the sample size sufficient to ensure adequate power to address the hypothesis being tested?

-Were correct statistical analysis used to support conclusions?

-Are there concerns about ethical or regulatory requirements being met?

Reviewer #1: NO

Reviewer #2: Please see summary and general comments.

Reviewer #3: I don’t find any reason to require additional analysis for acceptance of the manuscript.

**Results**

-Does the analysis presented match the analysis plan?

-Are the results clearly and completely presented?

-Are the figures (Tables, Images) of sufficient quality for clarity?

Reviewer #1: NO

Reviewer #2: Please see summary and general comments.

Reviewer #3: No major revisions on Result section, but there are several minor comments as described at Editorial and Data Presentation Modification.

**Conclusions**

-Are the conclusions supported by the data presented?

-Are the limitations of analysis clearly described?

-Do the authors discuss how these data can be helpful to advance our understanding of the topic under study?

-Is public health relevance addressed?

Reviewer #1: YES PARTIALLY

Reviewer #2: Please see summary and general comments.

Reviewer #3: No major revisions on Discussion section, but there are several minor comments as described at Editorial and Data Presentation Modification.

**Editorial and Data Presentation Modifications?**

Reviewer #1: (No Response)

Reviewer #2: Please see summary and general comments.

Reviewer #3: 1. Line 63 – References are needed on this sentence such as Platonov et al. 2011 and many others. 

2. Line 64-65 – Review articles were cited. Please cite original research articles such as Scoles et al. 2001 and Breuner et al. 2017 for I. scapularis, van Duijvendijk et al. 2016 for I. ricinus.

3. Line 65, 68 – The genus name ‘Borrelia’ was abbreviated at the beginning of the sentences here and in line 333 and 373, the genus name ‘Borrelia’ was spelled out at the beginning of the sentences. Please be consistent. I suggest spelling out the genus name when it was used at the beginning of the sentence. 

4. Line 67, 68 – again the references, please cite original research articles throughout the manuscript.

5. Line 40 – Through the manuscript, all the infection rates of Borrelia spp. in ticks were estimated from pooled samples. They should be converted to the infection rates of individual tick level such as Minimal Infection Rate (MIR) or other appropriate units otherwise the reported infection rates could be inflated as if all the ticks in the positive pool are infected.

6. Line 68 – The cited reference on this sentence is review paper. The original article that reported 3 distinct phylogenetic clades of B. miyamotoi should be cited here. (Cosson et al. 2014)

7. Line 72 – I don’t think that Fukunaga et al. 1995 and Scoles et al. 2001 confirmed that Apodemus argenteus and Peromyscus leucopus are the reservoir host of B. miyamotoi. Please provide better reference to support this sentence if the authors really think that those two rodent species were confirmed as the reservoir hosts of B. miyamotoi, or rephrase the sentence to explain the reservoir host status of those rodent species as described in the cited references. In addition, if the authors want to report any evidence regarding the reservoir hosts of B. miyamotoi, then I suggest adding the description or definition of reservoir host of tick-borne pathogen in the manuscript as described by Kahl et al. 2002.

8. Line 77 – What facts in the sentence are supported by Ref.10?

9. Line 118 – Change the subtitle of this paragraph since it also includes trapping methods.

10. Line 126 – What types of traps were used? Sherman trap? It might be helpful to better understand the sampling process if the authors provide more details of field sampling such as what types of trap? how many traps used? How the placement of the traps were arranged?

11. Line 140, 145, 148 – typo on the symbol of degree-Celsius. Remove space between degree symbol and Celsius symbol. 

12. Line 218 – Personally, I was a bit surprised from the result that only 43 ticks were collected from 640 captured animals and even there was no Ixodes larva. This result was not consistent from the previous report of the authors’ group, Takhampunya et al. 2021. Author’s opinion or explanation about this small number of ticks collected from captured animals and absence of Ixodes larva may help the readers to better understand the disease ecological feature of the vector ticks for B. miyamotoi in Thailand.

13. Line 242 – Since there are large variation on the sample sizes among animal species, I suggest adding confidence interval on each infection rate and seroprevalence rate.

14. Line 245 – It would be nice if the authors explain whether the positive ticks were collected from positive animals or not.

15. Line 252, Table 1. Typo. ‘Rattus exultans’ should be ‘Rattus exulans’. Please describe what animal species are in ‘Others’ category from trapped animals.

16. Line 337 – Again, the authors should use the term ‘Reservoir host’ more carefully. The cited reference (Ref. 10) just reported finding of B. miyamotoi DNA from the animal species listed in the sentence. This finding doesn’t prove that the positive animal species are ‘reservoir host’ of B. miyamotoi. They could be ‘potential’ reservoir hosts. Please rephrase the sentence.

17. Line 340 – This study didn’t find B. miyamotoi infection in humans as mentioned in line 346. This sentence can be deleted since the authors expressed same meaning at the sentence in line 349 – 351. 

18. Line 341 – Same as ‘reservoir host’, this study just found I. granulatus infected with B. miyamotoi collected from wild animals. It doesn’t prove Ixodes granulatus as the vector for B. miyamotoi. Please rephrase the sentence.

19. Line 375 – typo. ‘areassuch as’

20. Line 385 – Again, the detection of the bacteria from the animal species and ticks didn’t prove the animals or the ticks are ‘reservoir’ or ‘vector’, respectively. Please rephrase the sentence.

21. Line 390 – If there is a reference that reported the vector competence of Ixodes granulatus for B. miyamotoi, please cite the reference here. Otherwise, vector competence of I. granulatus for B. miyamotoi should be investigated first.

22. Line 392 – Please cite the reference that supports the high density of I. granulatus in mountainous areas.

**Summary and General Comments**

Reviewer #1: General comments

Major comments

The study is complex and aimed to evaluate the borrelia spp. species in a one health approach. The manuscript in general is well written and structured. However, some aspects are omitted and rise important questions/concerns. Few are presented below:

1. It is not clear if/how the sensibility and more important the specificity of the in-house ELISA assay was evaluated;

2. would be interesting to present the associations and or agreements between some results, for example:

a) if positive ticks were collected from positive hosts (same pathogen involved)?

b) there were cases of PCR positive and seropositive rodent individuals?

3. at some point is specified that also ticks from other animals were collected? why did not differentiated the results by tick’s vertebrate host? it is any statistically difference?

4. The statistical analysis is missing;

Minor comments

Avoid personal nouns and abbreviation at the beginning of the sentence (through all the manuscript). 

Specific comments:

line 47: In addition, seroprevalence of B. miyamotoi in human samples received from Phop Phra hospital, Tak province and rodents captured from Phop Phra was evaluated using a district using an in-house, direct enzyme-linked immunosorbent assay (ELISA) assay with B. miyamotoi glycerophosphodiester-phosphodiesterase (GlpQ) recombinant protein as coated antigen.

Reviewer #2: General Summary: Understanding geographic range, transmission cycle, comparative prevalence, genetic diversity (or lack thereof) and characterizing human infections of Borrelia miyamotoi are all topics worthy of research and publication. In this light, I read this manuscript that focuses on B. miyamotoi surveillance in Thailand with interest. This team of researchers in Thailand have trapped several genera of rodents (total animals n=640) from a one province (Tak Province), removed ticks off these rodents and tested both rodent tissue (spleen and kidney) and ticks for B. miyamotoi by PCR. In addition, they developed a rudimentary serological assay and tested sera collected from a blinded panel of febrile hospitalized patients. 

All said, while the topic is interesting, the methodology used for this study has serious errors and thus the results and conclusions are at best, not convincing and at worst, are irresponsible– thus I recommend this manuscript not be accepted for publication in its current form. Of most concern is the serological assay developed to test human samples. The description of this assay did not list the controls used (it would be recommended to develop using sera from patients who were positive for many different relapsing febrile illnesses (e.g., Borrelia hermsii, B. duttoni…as well as from patients positive for Borrelia spp. illnesses (e.g., Borrelia burgdorferi or Borrelia mayonii). Also, since these patients had acute infections as they were in the hospital, it would be appropriate to test by PCR in the case they hadn’t yet been able to mount an immune reaction. Note: the authors did test the patients sera by PCR and all sera samples tested negative but this result was really hidden in the manuscript. Secondly, to validate a serological assay, patients should have been screened for history of tick bite as well as relapsing fever and presented onset date and sera collection date (acute and chronic sera collection). At this point, it is irresponsible to state a 17.9% seroprevalence in humans in this study population using this non-validated serological assay. I would remove any mention of human testing from future papers unless this work is revised and all testing conducted appropriately for development of a human diagnostic assay with reporting of sensitivity and specificity.

The rodent and tick surveillance portion of this study also have serious errors and assumptions, most glaring is testing ticks removed from animals and assuming that these are infected with B. miyamotoi or act as possible vectors. While these ticks may harbor B.miyamotoi-infected blood acquired from their host, they may not able to sustain infection or transmit. To characterize the prevalence in ticks, unfed ticks must be collected from the environment prior to testing.

Minor issues that should be addressed include the definition of reservoir and vector: just finding a rodent positive does not make it clear it is a reservoir and finding a tick off an animal does not define the species as a vector: vector and host competence studies need to be conducted first. Nevertheless, it is a great first step in clarifying the transmission cycle in a region and I hope these researchers continue to move forward with additional studies on Borrelia miyamotoi in Thailand.

Reviewer #3: The manuscript describes informative findings regarding the eco-epidemiological feature of B. miyamotoi in Thailand. This report will improve our knowledge about the disease ecology and geographical distribution of B. miyamotoi. I appreciate that the authors provide serological evidence of B. miyamotoi infection in humans. The presentation needs improvement. The referencing is weak, missing several key references.

PLOS authors have the option to publish the peer review history of their article (what does this mean?). If published, this will include your full peer review and any attached files.

Reviewer #1: No

Reviewer #2: No

Reviewer #3: No
---

## [Decision Letter · Decision Letter 1]

3 Feb 2023

Dear Dr. Takhampunya,

Thank you very much for submitting your manuscript "Borrelia miyamotoi a neglected tick-borne relapsing fever spirochete in Thailand." for consideration at PLOS Neglected Tropical Diseases. As with all papers reviewed by the journal, your manuscript was reviewed by members of the editorial board and by several independent reviewers. The reviewers appreciated the attention to an important topic. Based on the reviews, we are likely to accept this manuscript for publication, providing that you modify the manuscript according to the review recommendations. 

Sincerely,

Wen-Ping Guo

Academic Editor

Mathieu Picardeau

Section Editor

Reviewer's Responses to Questions

**Key Review Criteria Required for Acceptance?**

**Methods**

-Are the objectives of the study clearly articulated with a clear testable hypothesis stated?

-Is the study design appropriate to address the stated objectives?

-Is the population clearly described and appropriate for the hypothesis being tested?

-Is the sample size sufficient to ensure adequate power to address the hypothesis being tested?

-Were correct statistical analysis used to support conclusions?

-Are there concerns about ethical or regulatory requirements being met?

Reviewer #3: No major additional analysis is required, but a few minor comments are suggested in Editorial and Data Presentation Modifications.

**Results**

-Does the analysis presented match the analysis plan?

-Are the results clearly and completely presented?

-Are the figures (Tables, Images) of sufficient quality for clarity?

Reviewer #3: No major revision is required, but a few minor comments are suggested in Editorial and Data Presentation Modifications.

**Conclusions**

-Are the conclusions supported by the data presented?

-Are the limitations of analysis clearly described?

-Do the authors discuss how these data can be helpful to advance our understanding of the topic under study?

-Is public health relevance addressed?

Reviewer #3: No major revision.

**Editorial and Data Presentation Modifications?**

Reviewer #3: 1. Sampling years for rodent trapping and human blood are confusing. Sampling periods were described differently in several places as listed below.

In introduction P4 Line 31-32, sampling years were described as “(2019-2020)” for ticks and rodents and “(2018-2019)” for human patients.

But in Materials and Methods, P5 Line 15-16, human sampling period was described as February-November 2017. Line 25, sampling period of rodent and ticks was described as "2019 (Feb thru Dec)".

In Results section, in the first subtitle (P9 Line 7-8) and the first paragraph, sampling year for rodents and ticks was described as 2019 only. 

In the second subtitle of the results and the contents, P10 Line 1-2 and Line 5-6, it looked like sampling year for human, rodents and ticks were 2019-2020.

P18 Line 4-5, human sampling period was described as "A total of 84 patient sera were collected from this hospital during 2018-2019."

S1 Table, there is a note describing "Samples (human) were collected during February-November 2017."

Please clarify the sampling years for rodents and humans, and be consistent throughout the manuscript.

2. P9 Line 1-4, this is a very poor sentence. It didn’t describe what data were estimated with the statistical method described in the sentence. 

The infection prevalence of ticks was estimated from pooled samples, so it was suggested to use any statistical methods for estimating infection rates from pooled samples. Because if not, the infection rates of ticks could be mis-interpreted as if all the ticks in the positive pools were infected. For example, in Abstract at P2, the authors summarized that they collected 43 ticks (line 5) and reported the infection rate of ticks as 22.2% (line 8). This could be misread as about 22.2% of 43 ticks (10 ticks) were infected. But if we use MIR (minimum infection rate) with the result of 6 positive pools from 27 pools of 43 ticks, the MIR of the ticks will be about 13.95% (6 positive ticks from 43 ticks).

The authors responded that they used the Excel add-in program for mosquito surveillance provided by CDC. I am glad that the author used MIR with the CDC program that doesn’t need an assumption required in the traditional MIR such as when a pool is positive then only one individual in that pool is positive. To my knowledge, the excel add-in tool will use the bias-corrected MLE for the point estimate infection rate and the skewness-corrected score for CI. The author can just use the point estimation of MIR calculated by bias-corrected MLE and the corrected score CI provided by the add-in tool. There is no need to add a column for MLE limits in the table. The table 1 in the revised manuscript still has the infection rate of ticks calculated by using the number of pools as the denominator. I still think that the infection rate of ticks should be reported from the individual tick number base, not from the pooled group testing.

**Summary and General Comments**

Reviewer #3: The manuscript describes informative findings regarding the eco-epidemiological feature of B. miyamotoi in Thailand. This report will improve our knowledge about the disease ecology and geographical distribution of B. miyamotoi. 

In the revised manuscript, the referencing and the structure of the manuscript were improved, but it still requires a few minor modifications as described at Editorial and Data Presentation Modifications. The authors already have an appropriate tool to re-estimate infection prevalence of ticks from pooled samples, so I believe that the suggestion on reporting infection prevalence of ticks can be done by minor revision.

PLOS authors have the option to publish the peer review history of their article (what does this mean?). If published, this will include your full peer review and any attached files.

Reviewer #3: No

Figure Files:

Data Requirements:

Reproducibility:

References

---

## [Editor Report · Decision Letter 2]

10 Feb 2023

Dear Dr. Takhampunya,

We are pleased to inform you that your manuscript 'Borrelia miyamotoi a neglected tick-borne relapsing fever spirochete in Thailand.' has been provisionally accepted for publication in PLOS Neglected Tropical Diseases.

Best regards,

Wen-Ping Guo

Academic Editor

Mathieu Picardeau

Section Editor

---

## [Editor Report · Acceptance letter]

16 Feb 2023

Dear Dr. Takhampunya,

We are delighted to inform you that your manuscript, "Borrelia miyamotoi a neglected tick-borne relapsing fever spirochete in Thailand.," has been formally accepted for publication in PLOS Neglected Tropical Diseases.

Best regards,

Shaden Kamhawi

co-Editor-in-Chief

Paul Brindley

co-Editor-in-Chief
